# Towards Multi-Table Learning: A Novel Paradigm for Complementarity Quantification and Integration

**Junyu Zhang**[†]  **Lizhong Ding**[†*]  **Minghong Zhang**  **Ye Yuan**  **Xingcan Li**
**Pengqi Li**  **Tihang Xi**  **Guoren Wang**  **Changsheng Li**
Beijing Institute of Technology
{zhangjunyu, lxc, pengqi.li, tihangxi, yuan-ye, lcs}@bit.edu.cn
lizhong.ding@outlook.com, zhangmh02@126.com, wanggrbit@126.com

## Abstract

Multi-table data integrate various entities and attributes, with potential interconnections between them. However, existing tabular learning methods often struggle to describe and leverage the underlying complementarity across distinct tables. To address this limitation, we propose the first unified paradigm for multi-table learning that systematically quantifies and integrates complementary information across tables. Specifically, we introduce a metric called complementarity strength (CS), which captures inter-table complementarity by incorporating relevance, similarity, and informativeness. For the first time, we systematically formulate the paradigm towards multi-table learning by establishing formal definitions of tasks and loss functions. Correspondingly, we present a network for multi-table learning that combines Adaptive Table encoder and Cross table Attention mechanism (ATCA-Net), achieving the simultaneous integration of complementary information from distinct tables. Experiments show that ATCA-Net effectively leverages complementary information and that the CS metric accurately quantifies the richness of complementarity across multiple tables. To the best of our knowledge, this is the first work to establish theoretical and practical foundations for multi-table learning.

## 1 Introduction

Multiple tables often exhibit inherent connections due to the interrelated nature of the entities they describe, which can yield valuable complementary information when such associations exist [15, 36]. These data have widespread applications across various domains, such as finance [19, 37], healthcare [29, 26] and geography [32, 39]. However, most existing table learning methods [38, 20, 21, 4] are limited to utilizing information from single or pairwise tables to accomplish tasks such as table completion [21, 26] or entity matching [23, 28]. There is a notable lack of methodologies to analysis and learn from multiple tables [18, 42].

Current approaches for analyzing inter-table relationships predominantly depend on similarity metrics [15, 36, 25, 16, 30]. However, these methods face significant challenges in quantifying meaningful complementary information between tables: (1) High similarity scores may originate from redundant tables (i.e., duplicated schemas) that provide negligible complementary value, while (2) completely unrelated tables inherently lack meaningful information complementarity. This distinction highlights the need for a new metric that quantifies complementarity by effectively balancing table similarity and relevance, thereby enabling a more comprehensive evaluation for multiple tables. On the other hand, to effectively harness the complementary information across multiple tables, two critical

---

[†]Equal contribution.
[*]Corresponding author.

technical barriers must be addressed. First, the inherent intra-table heterogeneity (divergent attribute semantics within tables) and inter-table heterogeneity (schema mismatch across tables) demand unified representation learning that preserves both table-specific characteristics and cross-table consistency. While modern single-table encoders [21, 38] excel at isolated table processing, they fundamentally lack cross-table alignment mechanisms. Second, current cross-table information fusion techniques either rely on explicit relational graphs [8, 18, 17] or employ computationally intensive full-table encoding via language models [32], neither of which efficiently captures latent complementary patterns. These challenges, including the inability to effectively describe and quantify multiple tables complementarity, as well as the difficulty in learning a unified representation and integrating complementary information across tables, underscore the necessity for a novel multiple tables learning paradigm that specifically designed to address these limitations.

To address these challenges in learning multiple tables, we propose a comprehensive paradigm that systematically quantifies and integrates complementary information. Specifically, we propose a novel metric termed complementarity strength (CS), which balances the statistical correlations and semantic similarities to quantify the complementarity between tables. A higher CS score indicates stronger inter-table correlation with minimal redundancy, overcoming the limitation of existing methods that exclusively prioritize similarity [36, 30] while neglecting complementarity. We develop the systematic, formal definitions for multiple tables learning tasks and their corresponding loss functions, offering clear guidance for model design and training processes. Furthermore, we propose a novel network for learning multiple tables, which consist of the adaptive table encoders and cross-tabular attention layers (ATCA-Net), effectively integrating complementary information from multiple tables. Extensive experiments demonstrate that CS effectively quantifies complementarity, while ATCA-Net efficiently integrates complementary information across tables. Our work provides both a solid theoretical foundation and practical guidelines for multiple table learning. The main contributions are summarized as follows:

- To the best of our knowledge, we are the first to define the complementarity strength (CS) metric, which quantifies the complementary information between multiple tables.

- We present the first comprehensive, systematically formalized definitions for multi-table learning tasks and their corresponding loss functions.

- We propose the new architecture, ATCA-Net, which efficiently processes multiple tables and integrates complementary information across tables.

## 2   Related work

**Tabular Representation Learning.**   Previous methods for learning tabular data mainly rely on information from a single table [9, 10, 11, 12, 13, 14]. Multi-table learning confronts substantial challenges in achieving semantic alignment and effectively extracting and integrating information across tables. These challenges encompass tackling schema heterogeneity [26, 43], resolving column-level semantic inconsistencies [2], and developing robust representations for tabular data [38, 37, 35]. Techniques such as learnable cell-level semantic encoding [21] and BERT-based embeddings [24] have facilitated column alignment,while table pretraining strategies [34], reconstruction tasks [35], and mutual information learning [38] underpin the development of table learning encoders. Despite these advancements, the extraction of complementary information across diverse tables remains an unresolved challenge. BERT-based approaches [32] have explored transforming tables into textual formats, and GNN methods [17] have enabled information exchange through foreign key linkages. In multimodal learning contexts [41], mechanisms such as cross-attention [1] or learnable query modules [27] are utilized for information fusion.However, these methods have not fully explored the information fusion in multiple tables settings.

**Multiple Tables Measurements.** Quantifying the strength of complementarity among multiple tables is crucial for advancing multi-table learning.Present research primarily focuses on similarity measurements, such as through column-level similarity evaluations to ascertain whether tables share analogous schemas [36, 40, 30]. Column-level similarity is typically assessed through the overlap of column names and values [31, 25] or via the matching of learned column embeddings [15]. For numerical columns, comparing whether they originate from similar distributions is an optional approach [3, 33]. These methodologies are extensively employed in applications like table augmentation [6] and dataset discovery within data lakes [5]. However, similarity metrics fail to reflect

the complementarity strength across tables,as identical tables, despite exhibiting maximum similarity, offer redundant information, while completely unrelated tables contribute minimally to the learning process. Consequently, the development of a novel metric for quantifying the complementarity of multiple tables is imperative.

## 3    Complementarity Strength

When tables exhibit relevance, they may provide complementary information that has the potential to enhance learning performance in multi-table integration. In contrast, unrelated tables tend to yield fragmented information that inherently lacks synergistic capacity. Beyond relevance alone, the degree of complementarity is also influenced by two additional factors: the informativeness of each table and the similarity between them. On the one hand, richer table content contributes more complementary information. On the other hand, higher similarity implies greater redundancy—when two tables are identical, despite their strong relevance, they offer no additional complementary benefit. Taken together, these analyses suggest that complementary information should be positively associated with both relevance and informativeness, but inversely associated with similarity. In this section, we formally define three key metrics: **relevance**, **similarity**, and **informativeness**, and derive a composite measure of inter-table **complementarity strength**.

### 3.1    Definition of Complementarity Strength

A multi-table collection is defined as $D := \{T^1, T^2, ..., T^K\}$, where the $k$-th table is given by $T^k := [v_{ij}^k]_{i \in [M_k], j \in [N_k]}, k \in [K]$, with $M_k$(resp. $N_k$) denoting the number of rows (resp. columns) in table $T_k$. Here, $v_{ij}^k$ represents the value of the $(i, j)$-th cell in table $T^k$. We denote the $i$-th row and $j$-th column of $T^k$ as $r_i^k = \{v_{ij}^k\}_{j \in [N_k]}$, $h_j^k = \{v_{ij}^k\}_{i \in [M_k]}$.

We begin by defining the **column-level similarity** between two tables. Each table $T^k$ is encoded by a pretrained model [15], producing a vector representation $z_j^k$ for each column $h_j^k$. [3] Given two column embeddings $z_j^k$ and $z_g^l$ from tables $T^k$ and $T^l$, their similarity is computed as the cosine similarity:

$$S_c(z_j^k, z_g^l) = \left| \frac{z_j^k \cdot z_g^l}{\|z_j^k\| \cdot \|z_g^l\|} \right| \in [0, 1], \tag{1}$$

where $\|\cdot\|$ denotes the Euclidean ($L_2$) norm. A similarity score of 1 indicates that the two columns are directionally aligned (i.e., linearly dependent), whereas a score of 0 indicates complete orthogonality, implying no semantic overlap.

Table-level relevance between two tables $T^k$ and $T^l$ is defined as the maximum similarity between any pair of columns across the two tables. Formally, the relevance score is computed as:

$$R_t(T^k, T^l) = \max_{j \in [N_k],\, g \in [N_l]} \left\{ S_c(z_j^k, z_g^l) \right\} \in [0, 1]. \tag{2}$$

A high relevance score indicates that at least one column pair is semantically similar, suggesting a possible relationship between the tables. Conversely, a low score implies that all column pairs are dissimilar, and the tables are considered unrelated.

Table-level similarity between two tables is computed by identifying the optimal bipartite matching [15] that maximizes the sum of column similarities. Each column from one table can be paired with at most one column from the other table or remain unpaired. Formally, table-level similarity between two tables $T^k$ and $T^l$ is defined as:

$$S_t(T^k, T^l) = \max_{U \in \mathcal{U}} \left\{ \sum_{(j,g) \in U} S_c(z_j^k, z_g^l) \right\} \in [0, \min(N_k, N_l)], \tag{3}$$

where $\mathcal{U}$ represents the set of all possible matchings between columns of $T^k$ and $T^l$.

We define the metric **informativeness** to quantify the richness of information within a table. A table is considered more informative when its columns convey diverse and non-redundant semantics, and less

---

[3]We adopt the pretrained model [15], which is trained on a large-scale corpus of 50M tables to learn generalizable column representations.

informative when its columns are similar. Formally, the informativeness of a table $T^k$ is computed as:

$$\text{Info}(T^k) = \sum_{i \in [N_k]} \left(1 - \frac{1}{N_k - 1} \sum_{j \in [N_k], j \neq i} \text{S}_\text{c}(z_i^k, z_j^k)\right) \in [0, N_k], \qquad (4)$$

where $\text{S}_\text{c}(z_i^k, z_j^k)$ denotes the similarity between the $i$-th and $j$-th columns of $T^k$. The inner summation quantifies how similar each column is to the rest of the table. By subtracting this value from 1, we obtain a measure of the uniqueness of the column. The overall $\text{Info}(T^k)$ score aggregates these uniqueness values in all columns, reflecting the total amount of distinct information the table contains.

Based on the above defined metrics, relevance ($\text{R}_\text{t}$), similarity ($\text{S}_\text{t}$) and informativeness (Info), we introduce the metric of **complementarity strength (CS)**, which quantifies how much novel and relevant information table $T^k$ contributes to $T^l$. It balances relevance against redundancy, and is formally defined as:

$$\text{CS}_\text{t}(T^k, T^l) = \text{Info}(T^k) \cdot \frac{\alpha \cdot \text{R}_\text{t}(T^k, T^l) \cdot \left(1 - \frac{1}{N_l}\text{S}_\text{t}(T^k, T^l)\right)}{1 + \frac{\gamma}{N_l}\text{S}_\text{t}(T^k, T^l)} \in [0, N_k], \qquad (5)$$

In our experiments, we set both $\alpha$ and $\gamma$ to 1. This formulation encourages high relevance and source table informativeness while penalizing redundancy by suppressing similarity. Notably, the complementarity is asymmetric, i.e., $\text{CS}(T^k, T^l) \neq \text{CS}(T^l, T^k)$. For example, if all columns in $T^l$ are already present in $T^k$, then $T^l$ contributes little new information to $T^k$. Conversely, since $T^k$ contains information not present in $T^l$, it can still provide complementary information to $T^l$.

Given a group of tables $\mathcal{D} = \{T^1, T^2, \ldots, T^K\}$, the aggregate complementarity strength from all other tables $D \setminus T^l$ to table $T^l$ is computed as follows:

$$\text{CS}_\text{g}(D \setminus T^l, T^l) = \sum_{T^k \in D \setminus T^l} \text{CS}_\text{t}(T^k, T^l) \in \left[0, \sum_{k=1}^{K} N_k\right].$$

## 3.2 Discussion of Complementarity Strength

Table 1 shows how the complementarity strength obtained for the same target table $T^c$ varies across five distinct groups of synthesized tables. All tables are randomly sampled by columns from the blastchar dataset from the OpenML Repository, with each table containing 5 columns. The $i$-th group is denoted as $\text{GP}_i = \{T^c, T_i^{a(j)}\}_{j \in [3]}$, including the target table $T^c$ and three auxiliary tables $T_i^{a(j)}$. Each group features a different column overlap ratio between $T^c$ and its auxiliary tables, ranging from 0 to 1 across groups $\text{GP}_1$ to $\text{GP}_5$. As anticipated, when the overlap is zero, the auxiliary tables struggle to form meaningful associations with the target table $T^c$, resulting in low relevance ($\text{R}_\text{t}$) and complementarity strength (CS). Conversely, when the overlap is high, the auxiliary tables become largely redundant, leading to increased similarity ($\text{S}_\text{t}$) but diminished complementarity. In cases of moderate overlap, the auxiliary tables establish some relevant associations with $T^c$, while maintaining relatively low redundancy, thus preserving a higher level of complementarity. The metrics we propose effectively capture and reflect these patterns.

Table 1: Complementarity Strength for target table $T^c$ across five table groups sampled from the blastchar dataset. Each group $\text{GP}_i$ has a different column-overlap ratio (0.0–1.0) between $T^c$ and three auxiliary tables $T_i^{a(j)}$. The metrics $\text{S}_\text{t}$, $\text{R}_\text{t}$, Info, and $\text{CS}_\text{t}$ report the mean $\pm$ standard deviation across the three auxiliary tables.

| Metric | **0.0** ($\text{GP}_1$) | **0.2** ($\text{GP}_2$) | **0.5** ($\text{GP}_3$) | **0.8** ($\text{GP}_4$) | **1.0** ($\text{GP}_5$) |
|---|---|---|---|---|---|
| $\text{S}_\text{t}(T_i^{a(j)}, T^c)$ | $2.81 \pm 0.17$ | $2.96 \pm 0.08$ | $3.15 \pm 0.13$ | $3.15 \pm 0.21$ | $3.58 \pm 0.34$ |
| $\text{R}_\text{t}(T_i^{a(j)}, T^c)$ | $0.78 \pm 0.05$ | $0.82 \pm 0.02$ | $0.89 \pm 0.01$ | $0.91 \pm 0.03$ | $0.95 \pm 0.04$ |
| $\text{Info}(T_i^{a(j)})$ | $2.10 \pm 0.34$ | $1.87 \pm 0.11$ | $2.13 \pm 0.50$ | $2.02 \pm 0.31$ | $1.90 \pm 0.03$ |
| $\text{CS}_\text{t}(T_i^{a(j)}, T^c)$ | $0.26 \pm 0.08$ | $0.28 \pm 0.02$ | $0.35 \pm 0.04$ | $0.29 \pm 0.07$ | $0.10 \pm 0.08$ |
| $\text{CS}_\text{g}(GP_i \setminus T^c, T^c)$ | $0.73$ | $0.83$ | $1.03$ | $0.87$ | $0.35$ |

# 4 Formulations of Multi-Table Learning

Traditional table learning paradigms were limited to utilizing information from a single table. In contrast, multi-table learning involves the integration of information from multiple tables. When complementary information exists between these tables, multi-table learning can leverage this complementarity to achieve performance improvements. In this section, we provide a formal definition of multi-table tasks along with the corresponding loss functions.

Let $T := [v_{ij}]_{i \in [M], j \in [N]}$ denote a table with $M$ rows (entities) and $N$ columns (attributes), where the $i$-th row and $j$-th column are represented by $r_i$ and $h_j$, respectively. In **single-table learning**, the objective is to minimize $\sum_{i=1}^{M} \mathcal{L}(f_\theta^s(\mathbf{r}_i), y_i)$, where $f_\theta^s(\cdot)$ is the prediction function for the single-table task, $y_i$ represents the ground truth, and $\mathcal{L}$ the loss function. In contrast, **multi-table learning** can leverage information from multiple tables. Let $D = \{T^c, T^{a(1)}, \ldots, T^{a(k)}\}$ denote a group of tables, where $T^c$ is the target table and $T^{a(1)}, \ldots, T^{a(k)}$ are auxiliary tables. The multi-table learning task can be formulated as:

$$\min_\theta \left( \sum_{i \in [M_c]} \mathcal{L} \left( f_\theta(r_i^c, T^{a(1)}, \ldots, T^{a(k)}), y_i^c \right) \right), \tag{6}$$

where $f_\theta(\cdot)$ is the prediction function that integrates information from the target table $T^c$ and the auxiliary tables $T_1^a, \ldots, T_k^a$.

In equation (6), $f_\theta$ implicitly learns the alignment and integration of complementary information across multiple tables. Consequently, we divide multi-table learning into two stages: pretraining for **cross-table alignment**, followed by training for the **integration of complementary information** across tables. The two pretraining tasks, multi-table reconstruction and cross-table correlation prediction, focus on learning the associations between attributes and entities, respectively, effectively aligning the complementary information across tables.

**Multi-table reconstruction** pretraining involves recovering the corrupted cells in tables. Given the corrupted tables $\hat{D} = \{\hat{T}^1, \hat{T}^2, \ldots, \hat{T}^K\}$, the learning target is to recovering the complement tables $D = \{T^1, T^2, \ldots, T^K\}$. By constructing $\hat{D}$ randomly, this task enables self-supervised training. We suppose that $\hat{T}^k \backslash (\hat{T}^k \cap T^k) = \{\hat{v}_{ij}^k\}_{(i,j) \in O^k}$ for the corrupted index set $O^k \subset R^k \times C^k$. This learning task can be formalized as:

$$\min_\theta \sum_{k \in [K]} \sum_{(i,j) \in O^k} \mathcal{L} \left( f_\theta \left( \hat{v}_{ij}^k; \hat{D} \right), v_{ij}^k \right). \tag{7}$$

**Cross-table correlation prediction** pretraining involves predicting the correlation between entities across different tables. For each pair of rows $r_i^k$ from table $T^k$ and $r_j^l$ from table $T^l$, the label $y_{ij}^{kl}$ represents the correlation between these two rows. The objective is to minimize the following loss:

$$\min_\theta \sum_{(T^k, T^l) \in D} \sum_{i \in [M_k], j \in [M_l]} \mathcal{L} \left( f_\theta(\bar{r}_i^k, \bar{r}_j^l; D), y_{ij}^{kl} \right). \tag{8}$$

Typically, the labels $y_{ij}^{kl}$ are constructed using pseudo-labeling techniques. Row sampling is performed on the vertical partition [37] of table, resulting in partial row $\bar{r}_i \subseteq r_i$. When the sampled rows $\bar{r}_i$ and $\bar{r}_j$ originate from the same row, the label $y_{ij}^{kl}$ is set to 1; otherwise, the label is assigned a value of 0.

After pretraining with the tasks of multi-table reconstruction and cross-table correlation prediction, the model gains the ability to align complementary information across tables. Building on this, the model integrates complementary information from multiple tables through the process defined in Equation (6). In Section 5, we propose a network architecture designed to implement this two-stage learning process, facilitating effective multi-table learning.

# 5 Architecture for Multi-table Learning

Multi-table learning involves aligning and integrating complementary information across tables; to achieve this, we introduce ATCA-Net, which combines an Adaptive Table encoder and a Cross-Table Attention mechanism. To align the complementary information, we introduce the adaptive table encoder, which enables adaptation to various table schemas. Through the multi-table pretraining strategy, we project distinct tables into a unified representation space and model the correlations

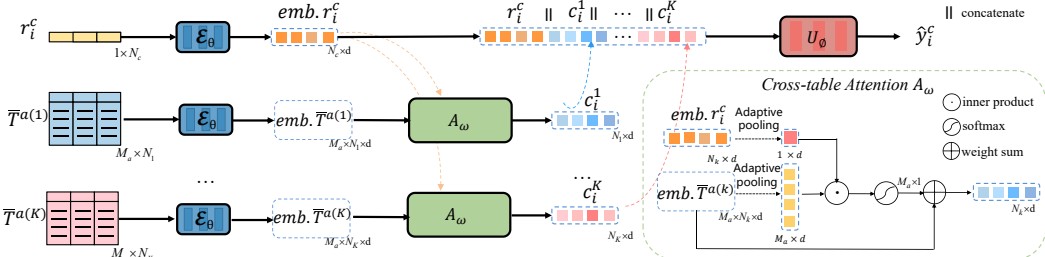

Figure 1: ATCA-Net operates in two stages: cross-table alignment stage and complementary information integration stage. In the first stage, the adaptive table encoder $\mathcal{E}_\theta$ is pretrained using Equation (9). In the second stage, the cross-table attention mechanism $\mathcal{A}_\omega$ extracts complementary information between tables and the fusion module $\mathcal{U}_\phi$ integrates the complementary information for prediction.

between cross-table attributes and entities. To integrate the complementary information, we propose a cross-table attention mechanism, which computes aggregation weights for auxiliary tables based on the target table's entities, ultimately achieving the fusion of complementary information across multiple tables.

## 5.1 Cross-table Alignment

We propose an adaptive table encoder with BERT-based [24] cell-level representation to handle various table schemas. Through multi-table reconstruction (Equation (7)) and cross-table correlation prediction (Equation (8)) pretraining, data from different tables are projected into a unified representation space, ensuring alignment of complementary information across tables.

**Cell-level representations** are obtained using a pretrained BERT model [24] to promote semantic consistency across heterogeneous attributes in multiple tables. Specifically, for each categorical cell, we concatenate the category value $v_{\mathrm{cat}}$ with its attribute name $a_{\mathrm{cat}}$ as "$v_{\mathrm{cat}\_}a_{\mathrm{cat}}$" and feed this string into BERT to produce a cell-level embedding. For each numerical cell, we encode the attribute name $a_{\mathrm{num}}$ with BERT and scale the resulting embedding by the normalized cell value to obtain the cell-level embedding. Finally, to standardize dimensionality, we apply average pooling to project each cell embedding to $\mathbb{R}^d$. Thus, an input table $T$ with $M$ rows and $N$ columns is represented as a tensor $X \in \mathbb{R}^{M \times N \times d}$. To enable row-level representations, we prepend a special [CLS] column to the table; the resulting tensor has shape $X \in \mathbb{R}^{M \times (N+1) \times d}$. Notably, we adopt mini-batch training, where each batch consists of an input sub-table sampled from the complete table with a fixed number of rows (e.g., $M = 64$).

Building upon these cell-level representations, the **adaptive encoder** $\mathcal{E}_\theta$ is designed to encode various tables using a two-dimensional Transformer [24] that captures dependencies along both the row and column axes. The encoder stacks Transformer layers that model contextual relationships in both dimensions. For table embeddings $\{\mathbf{x}_{ij}\}_{i=1\,j=1}^{M\quad N+1}$ with $\mathbf{x}_{ij} \in \mathbb{R}^d$, we apply a standard multi-head self-attention layer to the row sequence $(\mathbf{x}_{i1}, \ldots, \mathbf{x}_{i(N+1)})$ to obtain contextualized row outputs $(\mathbf{z}_{i1}^r, \ldots, \mathbf{z}_{i(N+1)}^r)$; likewise, we apply the same operation to the column sequence $(\mathbf{x}_{1j}, \ldots, \mathbf{x}_{Mj})$ to obtain column outputs $(\mathbf{z}_{1j}^c, \ldots, \mathbf{z}_{Mj}^c)$. The output representation of cell $(i, j)$ is the average of the two directional outputs, $\mathbf{z}_{ij} = \frac{1}{2}(\mathbf{z}_{ij}^r + \mathbf{z}_{ij}^c)$, which integrates context from both rows and columns.

We introduce two **multi-table pretraining** strategies for the adaptive encoder $\mathcal{E}_\theta$: multi-table reconstruction and cross-table correlation prediction. These strategies enable the encoder to align table representations across different tables. In the **multi-table reconstruction** pretraining, misalignment and masking [34] corruption strategies are randomly applied to various positions within the tables. Training pairs $\{(T_i, \hat{T}_i)\}_{i \in I}$ are randomly generated from the multiple tables $D$, where $T_i \in D$ is an original table and $\hat{T}_i$ is the corrupted version. Let $\tilde{T}_i$ represent the reconstructed table based on the encoded representation $\mathcal{E}_\theta(\hat{T}_i)$. The corresponding loss function for Equation (7) can be rewritten as:

$$\mathcal{L}_{\mathrm{rec}} = \sum\nolimits_{i \in [I]} \left( \lambda_{\mathrm{num}} \sum\nolimits_{\tilde{v} \in \tilde{T}_i^{\mathrm{num}}} \|\tilde{v} - v\|_2^2 + \lambda_{\mathrm{cls}} \sum\nolimits_{\tilde{v} \in \tilde{T}_i^{\mathrm{cls}}} \mathcal{H}(\tilde{v}, v) \right),$$

where $\tilde{v} \in \tilde{T}_i^{num}$ or $\tilde{v} \in \tilde{T}_i^{cls}$ denotes the values of numerical or categorical types, respectively. The term $\mathcal{H}$ represents the cross-entropy loss, and $v \in T_i$ is the ground truth of $\tilde{v}$.

In the pretraining task of **cross-table correlation prediction**, we randomly sample pairs of tables $T$ and $T^\dagger$ from the dataset $\mathcal{D}$ during each iteration $i \in [I]$. These tables are then vertically partitioned into sub-tables $P$ and $P^\dagger$[37]. Let $\bar{r}$ and $\bar{r}^\dagger$ represent rows from $P$ and $P^\dagger$, respectively. The goal of the cross-table correlation prediction task is to predict the correlation between rows in $P$ and $P^\dagger$. The corresponding loss function for Equation (8) can be rewritten as:

$$\mathcal{L}_{\text{cor}} = \frac{1}{|I|} \sum_{i \in [I]} \sum_{\bar{r} \in P_i,\, \bar{r}^\dagger \in P_i^\dagger} \mathcal{H}\left(\psi(\bar{r}, \bar{r}^\dagger), y_i\right),$$

where $\psi(\bar{r}, \bar{r}^\dagger)$ denotes the cosine similarity between the [CLS] embeddings of rows $\bar{r}$ and $\bar{r}^\dagger$, and $y_i = 1$ if they correspond to the same entity (otherwise $y_i = 0$). Let $\lambda_{\text{rec}}$ and $\lambda_{\text{cor}}$ denote the combination weights. The total pretraining loss is

$$\mathcal{L}_{\text{pre}} = \lambda_{\text{rec}}\, \mathcal{L}_{\text{rec}} + \lambda_{\text{cor}}\, \mathcal{L}_{\text{cor}}. \tag{9}$$

## 5.2 Complementary Information Integration

Let $\mathcal{D} = \{T^c, T^{a(1)}, T^{a(2)}, \ldots, T^{a(K)}\}$ denote a collection of tables, where $T^c$ is the target table and $T^{a(1)}, \ldots, T^{a(K)}$ are auxiliary tables. The adaptive encoder $\mathcal{E}_\theta$ is first pretrained across all tables in $D$ through the objective defined in Equation (9). We subsequently introduce the multi-table fusion module $\mathcal{F}_\xi$ to integrate complementary information across tables.

During the training of $\mathcal{F}_\xi$, each iteration involves sampling a row from the target table $T^c$ as the query instance, while constructing contextual sub-tables $\{\bar{T}^{a(k)}\}_{k \in [K]}$ by randomly selecting $M_a$ rows from each auxiliary table $T^{a(k)}$ to ensure computational tractability. The fusion process can be represented as

$$\mathcal{F}_\xi\left(\hat{r}_i^c, \{\hat{r}_s^{a(k)}\}_{k \in [K], s \in M_a}\right) = \mathcal{U}_\phi\left(\hat{r}_i^c, \mathcal{A}_\omega\left(\hat{r}_i^c, \{\hat{r}_s^{a(k)}\}_{k \in [K], s \in M_a}\right)\right) \tag{10}$$

where $\hat{r}_i^c \in \mathbb{R}^{(N_c+1) \times d}$ denotes the encoded embedding of the $i$-th entity in target table , $\hat{r}_s^{a(k)} \in \mathbb{R}^{(N_k+1) \times d}$ represents the $s$-th entity embedding of $k$-th auxiliary sub-table, while $\mathcal{A}_\omega$ and $\mathcal{U}_\phi$ denoting the cross-table attention module and the fusion module.

$\mathcal{A}_\omega$ leverages cross-table attention to extract and integrate complementary information. To enable effective interaction, adaptive pooling is first employed to project the row-level representations from different tables into a unified feature space. Let $\hat{r}_i^k = \{\hat{v}_{i1}^k, \hat{v}_{i2}^k, \ldots, \hat{v}_{iN_k}^k\}$ represent the encoded embedding of the $i$-th entity in table $\bar{T}^k$, where $N_k$ is the number of columns in table $T^k$. Through adaptive pooling, the entity $\hat{r}_i^k \in \mathbb{R}^{N_k \times d}$ is transformed into a vector $z_i^k \in \mathbb{R}^d$ by : $z_i^k = \sum_{j \in [N_k]} \hat{v}_{ij}^k / (1 + e^{-\langle W, \hat{v}_{ij}^k \rangle})$, where $W \in \mathbb{R}^d$ is the learnable weights.

Through aligning the row-level information from different tables into the $d$-dimensional space, we are able to further extract the complementary information between the $\hat{r}_i^c$ and auxiliary tables. Denote $c_i^k$ as the complementary information from table $\bar{T}^{a(k)}$ to $\hat{r}_i^c$. It can be fomulated as follows:

$$c_i^k = \sum_{j=1}^{M_a} \hat{r}_j^k \cdot \frac{e^{\langle z_i^c, z_j^k \rangle}}{\sum_{g=1}^{M_a} e^{\langle z_i^c, z_g^k \rangle}} \in \mathbb{R}^{N_k \times d}.$$

The complementary information from all auxiliary tables is concatenated and fed into the fusioner $\mathcal{U}_\phi$ to produce the final prediction: $\hat{y}_i^k = \mathcal{U}_\phi\left(\hat{r}_i^k \,\|\, c_i^1 \,\|\, c_i^2 \,\|\, \cdots \,\|\, c_i^K\right)$. Here, the fusioner $\mathcal{U}_\phi$ is implemented as a stack of standard Transformer layers. The learnable parameters $\xi$ of $\mathcal{F}_\xi$ comprise the parameters $\omega$ of $\mathcal{A}_\omega$ and the parameters $\phi$ of $\mathcal{U}_\phi$; all parameters are optimized according to the objective in Equation (6).

## 6 Experiment

In this section, we demonstrate ATCA-Net's ability to integrate complementary information across multiple tables and show that the CS metric effectively measures inter-table complementarity. We

Table 2: Statistics of real-world datasets, organized into three groups for multiple tables setting.

| Group | Name | Datapoints | Categorical | Numerical | Positive Ratio |
|---|---|---|---|---|---|
| | credit-g (CG) | 1,000 | 13 | 7 | 0.70 |
| | adult (AD) | 48,842 | 12 | 2 | 0.24 |
| **Group 1** | blastchar (BL) | 7,043 | 16 | 3 | 0.27 |
| | 1995-income (IC) | 32,561 | 8 | 6 | 0.24 |
| | employee churn(EC) | 1543 | 6 | 3 | 0.57 |
| | banking churn(BC) | 28382 | 4 | 15 | 0.18 |
| **Group 2** | churn modeling(CM) | 10000 | 5 | 8 | 0.20 |
| | customer churn(CC) | 64374 | 3 | 4 | 0.47 |
| **Group 3** | support (SP) | 9106 | 14 | 29 | 0.68 |
| | diabetes (DI) | 10000 | 19 | 3 | 0.13 |

evaluate performance on classification tasks and report the mean over five runs. Specifically, we construct three groups of real-world multi-table datasets (as shown in Table 2) and four groups of synthesized datasets; detailed settings for the datasets are provided in Appendix A.1.

To compute the complementarity coefficient across multiple tables, we set the hyper parameters $\alpha = 1$ and $\gamma = 1$ in Equation (5). We set the $\mathcal{L}_{pre}$ combination weights $\lambda_{rec} = 1$, and $\lambda_{cor} = 1$ in Equation (9). During training stage, we randomly sample sub-tables from the original tables by selecting rows and columns, with the number of rows fixed at 64 and the number of columns varying between 2 and the maximum number of columns. The sub-tables is firstly embedded by BERT, where each cell is represented as a 768-dimensional vector, which is then reduced to 192 dimensions via a fully connected layer to support larger table inputs. These embeddings are subsequently processed by a shared-weight adaptive encoder. Additional experimental details and supplementary results are provided in Appendix B.

## 6.1 Baselines

We include the following baselines for comparison: the methods learning from single table, including Logistic Regression (LR), XGBoost [7], Multilayer Perceptron (MLP), Saint [34], and FT-Transformer (FT-Trans) [21]; and the method learning from multiple tables TransTab [37]. To further validate the effectiveness of our approach, we extend FT-Transformer and Saint to multiple tables learning settings, through single-table encoder pretraining and multiple tables fusion to further validate the effectiveness of our approach.

## 6.2 Results

**Performance on real-world datasets**  We evaluate ATCA-Net's performance on classification tasks across three groups of tables, with AUC scores reported in Table 3. Most baseline methods [22, 21, 34, 7] are trained and tested on single tables. ATCA-Net and TransTab [37] employ cross-table collaborative training across grouped tables, The results demonstrate that ATCA-Net achieves state-of-the-art performance, outperforming all comparative methods. ATCA-Net (S) denotes the model variant trained and tested solely on single-table representations without any fusion or pretraining processes. ATCA-Net (M) represents the complete multi-table training pipeline with all fusion mechanisms. This set of comparisons also demonstrates the effectiveness of multi-table alignment pretraining and the integration mechanism of complementary information.

**Performance on synthetic datasets**  To further demonstrate the effectiveness of our approach in multiple tables learning tasks, we design the variants learned on multiple tables based on SAINT [34] and FT-Transformer [21]. Specifically, we train independent encoders on each table using SAINT and FT-Transformer, followed by a fusion network incorporating the cross-table attention mechanism described in Section 5.2. Furthermore, we evaluated TransTab [37], which employs a collaborative training mechanism in related tables to facilitate transfer learning. Furthermore, we implement an AdaBoost ensemble, where base decision tree learners are trained separately on individual tables.

Table 3: AUC performance on the real-world datsets. Except for ATCA-Net and Trans-tab trained on multiple tables, all methods are trained and tested on single table.

| Model | Group 1 | | | | Group 2 | | | | Group 3 | |
| | AD | BL | IC | CG | EC | BC | CM | CC | SP | DI |
|---|---|---|---|---|---|---|---|---|---|---|
| LR | 0.851 | 0.801 | 0.869 | 0.720 | 0.805 | 0.768 | 0.734 | 0.736 | 0.830 | 0.827 |
| XGBoost [7] | 0.912 | 0.821 | **0.925** | 0.726 | 0.786 | **0.817** | 0.837 | 0.792 | 0.851 | 0.832 |
| MLP | 0.904 | 0.832 | 0.892 | 0.643 | 0.769 | 0.776 | 0.692 | 0.782 | 0.773 | 0.754 |
| TabPFN v2 [22] | 0.878 | 0.837 | 0.888 | 0.718 | 0.753 | 0.809 | 0.817 | 0.756 | 0.820 | 0.828 |
| FT-Trans[21] | 0.827 | 0.756 | 0.832 | 0.674 | 0.714 | 0.758 | 0.713 | 0.738 | 0.769 | 0.678 |
| Saint[34] | 0.859 | 0.792 | 0.855 | 0.701 | 0.816 | 0.671 | 0.826 | 0.795 | 0.763 | 0.862 |
| Trans-tab[37] | 0.881 | 0.825 | 0.893 | 0.707 | 0.798 | 0.751 | 0.814 | 0.764 | **0.841** | 0.822 |
| ATCA-Net (S) | 0.911 | 0.825 | 0.911 | 0.617 | 0.812 | 0.724 | 0.836 | 0.757 | 0.801 | 0.805 |
| ATCA-Net (M) | **0.913** | **0.846** | 0.918 | **0.796** | **0.862** | 0.758 | **0.879** | **0.803** | 0.828 | **0.918** |

Table 4: AUC performance for the same target table on the blastchar synthetic dataset, trained with five different table groups. CS denotes the complementarity strength, indicating the degree of complementarity offered by each group.

| Methods | GP 1 | GP 2 | GP 3 | GP 4 | GP 5 |
|---|---|---|---|---|---|
| CS | 0.73 | 0.83 | 1.03 | 0.87 | 0.35 |
| Saint(S) | 0.653 | 0.653 | 0.653 | 0.653 | **0.653** |
| FT-Trans(S) | 0.557 | 0.557 | 0.557 | 0.557 | 0.557 |
| TransTab(S) | 0.599 | 0.599 | 0.599 | 0.599 | 0.599 |
| ATCA-Net(S) | 0.625 | 0.625 | 0.625 | 0.625 | 0.625 |
| Adaboost(M) | 0.613 | 0.601 | 0.617 | 0.609 | 0.620 |
| Saint(M) | 0.620 | 0.608 | 0.607 | 0.606 | 0.617 |
| FT-Trans(M) | 0.557 | 0.501 | 0.550 | 0.523 | 0.633 |
| TransTab(M) | 0.603 | 0.599 | 0.606 | 0.606 | 0.595 |
| ATCA-Net(M) | **0.657** | **0.654** | **0.661** | **0.663** | 0.625 |

The results across five multiple tables groups are present in Table 4. Our method consistently achieves the best performance. It is worth noting that, since the evaluation is conducted on the target table, the methods with single-table setting produce the same experimental results across all groups. Notably, apart from ATCA-Net, other multiple tables methods exhibited varying degrees of performance degradation compared to their single-table counterparts. This suggests that the introduction of noise makes it difficult to extract complementary information, likely due to the absence of a cross-table representation alignment mechanism. Our method, leveraging a two-stage framework for multiple tables representation learning and fusion, achieved the best results, with performance improving as the complementarity strength increases. We present additional experimental results on other synthetic datasets in the Appendix B.

## 6.3 Analysis of Complementarity Strength

In Section (3.2), we provided an initial discussion on the performance of complementary strength. Here, we further analyze it in detail. Using the same setup as Section (3.2), we constructed 4 synthetic table datasets. The details of the synthetic dataset and the complementary strength analysis are reported in Appendix B. Figure 2 presents the performance of the proposed multi-table learning method, ATCA-Net, along with the correlation analysis of complementary strength, showing the results across four datasets. The x-axis represents the complementarity strength, and the y-axis represents the AUC or ACC performance. A consistent trend is observed across all four datasets, where the performance of ATCA-Net gradually improves as the complementary strength increases. This demonstrates that the CS metric accurately measures the degree of complementarity between tables, while also validating the ability of ATCA-Net to extract and integrate complementary information from multiple tables.

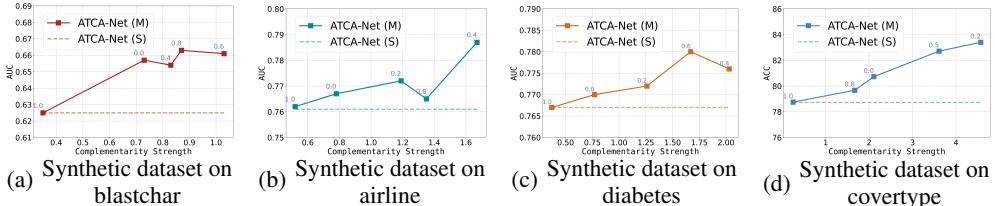

| (a) Synthetic dataset on blastchar | (b) Synthetic dataset on airline | (c) Synthetic dataset on diabetes | (d) Synthetic dataset on covertype |

Figure 2: Experimental results on the four synthetic datasets demonstrate that multi-table approaches consistently outperform single-table methods across all datasets. Notably, ATCA-Net exhibits progressively enhanced performance with increasing complementary strength between tables. The corresponding table coverage metrics are annotated adjacent to each point.

## 7  Conclusion

In summary, to address the limitation of existing table learning methods in capturing and utilizing the interrelationship among multiple tables, we systematically propose a novel paradigm for learning from multiple tables, which quantifies and integrates complementary information between tables. We introduce the metric complementarity strength to measure how a set of tables can contribute to a target table. Furthermore, we provide a formal definition for multiple table learning tasks, along with corresponding loss functions. We then present ATCA-Net, which incorporates a unified representation learning stage for multiple tables and a complementary integration stage, facilitating the integration of complementarity from multiple tables. Experiments on multiple tables entity prediction tasks demonstrate the effectiveness of our approach for learning multiple tables. This work explores the learning of multiple tables from both theoretical and practical perspectives, providing a significant foundation for the advancement of the field.

## Acknowledgments and Disclosure of Funding

This work was supported by the National Key Research and Development Program of China under Grant 2022YFB2703100, the Joint Funds of the National Natural Science Foundation of China under Grant U22A2099, the National Natural Science Foundation of China under Grant 62376028, the Excellent Young Scientists Fund (Overseas) of the National Natural Science Foundation of China, and the National Key Scientific Instruments and Equipment Development Project under Grant 62427808.

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

## A  Datasets and Implement Details

**Real-World Datasets**  In our experiments, we examine three distinct groups of datasets. As shown in Table 2, the first group includes four credit-related tables, encompassing data for tasks such as credit card risk prediction and adult income estimation. The second group consists of four tables focused on predicting both bank customer churn and employee attrition. The third group contains two tables centered on disease prediction, based on health indicators and lifestyle habits. In the same group of tables, there are no explicit entity links, but potential associations exist due to the shared similar attributes and entities.

**Synthetic Datasets**  The synthetic datasets were derived from the OpenML repository, including blastchar, airline, diabetes, and covertype, with details provided in Table 5. Each synthetic dataset was randomly partitioned by columns into multiple groups of tables; each group contains a shared target table and several auxiliary tables. Across groups, both the overlap ratio and the complementarity strength vary. The synthesis procedure is described in Appendix B.

Table 5: Statistics of synthetic datasets

| Name | Datapoints | Categorical | Numerical | Positive Ratio |
|------|-----------|-------------|-----------|----------------|
| blastchar (BL) | 7,043 | 16 | 3 | 0.27 |
| diabetes (DI) | 10,000 | 19 | 3 | 0.13 |
| airline (AIR) | 129,880 | 17 | 4 | 0.55 |
| covertype (COV) | 81,013 | 44 | 10 | - |

## B  Additional Results

**Synthetic dataset on airline**  Airline dataset has been used to construct five groups of tables by randomly sampling its columns. Each group contains 4 tables, and each table contains $N = 5$ columns. The $i$-th group is denoted as $\text{GP}_i = \{T^c, T_i^{a(j)}\}_{j \in [3]}$, including the target table $T^c$ and three auxiliary tables $T_i^{a(j)}$. Each group features a different column overlap ratio between $T^c$ and its auxiliary tables, ranging from 0 to 1 across groups $\text{GP}_1$ to $\text{GP}_5$.

Table 6 reports the correlation between CS and the column-overlap ratio on the airline synthetic dataset, exhibiting the same trend as in Section 3.2. Using the synthetic dataset on airline, we further report the performance of the same target table $T^c$ across different groups, as shown in Table 7. As CS increases, the proposed ATCA-Net consistently achieves better performance, a trend also illustrated in Figure 2(b).

Table 6: The correlation between CS and the overlap rate of the synthetic dataset on airline. Values are reported as mean ± standard deviation across the three auxiliary tables for each group.

| Metric | 0.0 ($GP_1$) | 0.2 ($GP_2$) | 0.5 ($GP_3$) | 0.8 ($GP_4$) | 1.0 ($GP_5$) |
|--------|-------------|-------------|-------------|-------------|-------------|
| $S_t(T^{a(j),T^c})$ | $1.70 \pm 0.10$ | $2.03 \pm 0.19$ | $2.48 \pm 0.07$ | $3.48 \pm 0.24$ | $3.82 \pm 0.07$ |
| $R_t(T^{a(j),T^c})$ | $0.37 \pm 0.02$ | $0.48 \pm 0.04$ | $0.65 \pm 0.04$ | $0.83 \pm 0.07$ | $0.88 \pm 0.01$ |
| $\text{Info}(T_i^{a(j)})$ | $1.41 \pm 0.28$ | $1.97 \pm 0.13$ | $2.54 \pm 0.27$ | $3.08 \pm 0.13$ | $2.31 \pm 0.03$ |
| $\text{CS}(T_i^{a(j),T^c})$ | $0.26 \pm 0.05$ | $0.40 \pm 0.03$ | $0.56 \pm 0.06$ | $0.45 \pm 0.05$ | $0.19 \pm 0.02$ |
| $\text{CS}(GP_i \setminus T^c, T^c)$ | 0.78 | 1.19 | 1.67 | 1.35 | 0.56 |

**Synthetic dataset on covertype**  All tables are randomly sampled by columns from the covertype dataset from the OpenML Repository, with each table containing 10 columns. The $i$-th group is denoted as $\text{GP}_i = \{T^c, T_i^{a(j)}\}_{j \in [3]}$, including the target table $T^c$ and three auxiliary tables $T_i^{a(j)}$. Each group features a different column overlap ratio between $T^c$ and its auxiliary tables, ranging from 0 to 1 across groups $\text{GP}_1$ to $\text{GP}_5$. An analysis of the correlation between overlap rate and CS metrics presented in Table 8. The corresponding performance metrics are provided in Table 9.

Table 7: AUC performance of synthetic dataset on airline

| Methods | 0.0 ($GP_1$) | 0.2 ($GP_2$) | 0.4 ($GP_3$) | 0.8 ($GP_4$) | 1.0 ($GP_5$) |
|---|---|---|---|---|---|
| CS | 0.78 | 1.19 | **1.67** | 1.35 | 0.56 |
| Tabpfn (S) | 0.753 | 0.753 | 0.753 | 0.753 | 0.753 |
| FT-Trans (S) | 0.691 | 0.691 | 0.691 | 0.691 | 0.691 |
| Saint (S) | 0.753 | 0.753 | 0.753 | 0.753 | 0.753 |
| TransTab (S) | 0.760 | 0.760 | 0.760 | 0.760 | 0.760 |
| ATCA-Net (S) | 0.742 | 0.742 | 0.742 | 0.742 | 0.742 |
| Adaboost (M) | 0.713 | 0.714 | 0.710 | 0.711 | 0.711 |
| TransTab (M) | 0.760 | 0.760 | 0.759 | 0.760 | **0.762** |
| ATCA-Net (M) | **0.767** | **0.772** | **0.787** | **0.765** | 0.745 |

Table 8: The correlation between CS and the overlap rate of the synthetic dataset on covertype. Values are reported as mean ± standard deviation across the three auxiliary tables for each group.

| Metric | 0.0 ($GP_1$) | 0.2 ($GP_2$) | 0.5 ($GP_3$) | 0.8 ($GP_4$) | 1.0 ($GP_5$) |
|---|---|---|---|---|---|
| $S_t(T_i^{a(j)}, T^c)$ | $2.00 \pm 0.09$ | $3.75 \pm 0.15$ | $6.02 \pm 0.12$ | $7.87 \pm 0.05$ | $9.53 \pm 0.11$ |
| $R_t(T_i^{a(j)}, T^c)$ | $0.27 \pm 0.02$ | $0.66 \pm 0.02$ | $0.86 \pm 0.04$ | $0.92 \pm 0.02$ | $0.98 \pm 0.00$ |
| $\text{Info}(T_i^{a(j)})$ | $3.89 \pm 0.45$ | $5.04 \pm 0.09$ | $5.60 \pm 0.15$ | $5.06 \pm 0.24$ | $3.47 \pm 0.03$ |
| $CS_t(T_i^{a(j)}, T^c)$ | $0.70 \pm 0.11$ | $1.52 \pm 0.07$ | $1.21 \pm 0.08$ | $0.55 \pm 0.02$ | $0.08 \pm 0.03$ |
| $CS_g(GP_i \setminus T^c, T^c)$ | 2.11 | 4.57 | 3.61 | 1.67 | 0.25 |

Table 9: ACC performance of covertype dataset

| Methods | 0.0($GP_1$) | 0.2($GP_2$) | 0.5($GP_3$) | 0.8($GP_4$) | 1.0($GP_5$) |
|---|---|---|---|---|---|
| CS | 2.11 | **4.57** | 3.61 | 1.67 | 0.25 |
| TabPFN v2 (S) | 72.65 | 72.65 | 72.65 | 72.65 | 72.65 |
| FT-Trans (S) | 78.39 | 78.39 | 78.39 | 78.39 | 78.39 |
| Saint (S) | 77.85 | 77.85 | 77.85 | 77.85 | 77.85 |
| ATCA-Net(S) | 78.73 | 78.73 | 78.73 | 78.73 | 78.73 |
| Adaboost (M) | 72.46 | 72.94 | 72.20 | 73.03 | 72.86 |
| **ATCA-Net (M)** | **80.73** | **82.39** | **83.40** | **79.67** | **78.75** |

# C Limitations

To our knowledge, we are the first to quantify complementarity across multiple tables and to introduce an architecture that extracts and integrates it. Several limitations remain: (i) CS depends on column representations from large-scale pretraining, making its accuracy sensitive to representation quality; (ii) CS is defined via column-level similarity and may not align with task-specific gains, motivating task-oriented measures of complementarity; and (iii) although CS correlates positively with predictive performance on real and synthetic data, the gains are modest, underscoring the need for architectures that more effectively exploit complementarity.

