# OpenReview forum: "Towards Multi-Table Learning: A Novel Paradigm for Complementarity Quantification and Integration"
_NeurIPS.cc/2025/Conference — NeurIPS 2025 spotlight_

### Official Review · Reviewer_MnRt · 2025-07-02

**Clarity:** 2
**Significance:** 2
**Originality:** 2
**Rating:** 4
**Confidence:** 3

**Summary:**

This paper proposes a metric, Complementarity Strength (CS), and a learning strategy for multi-table setting. CS blends relevance, similarity, and informativeness to quantify how much new, non‑redundant information one table adds to another. The authors formalizes self‑supervised pretraining objectives (multi‑table reconstruction and cross‑table correlation prediction) and implements them in ATCA‑Net, which uses an adaptive 2D‑Transformer encoder plus a cross‑table attention fusion module to align and integrate auxiliary tables into a target table. Experiments on real‑world and synthetic classification tasks show that CS predicts downstream gains and that ATCA‑Net outperforms both single‑table baselines and prior multi‑table methods.

**Questions:**

please see weaknesses.

**Ethical Concerns:**

["NO or VERY MINOR ethics concerns only"]

**Final Justification:**

After considering the rebuttal and other reviews, I recommend a score of 4 (Borderline Accept) for this paper. The authors have adequately addressed my concerns regarding hyperparameter selection and provided clarification on their impact through additional discussion. While I am not directly working in the area of table learning, I acknowledge the contributions highlighted by other reviewers, particularly the novelty of the proposed metric (Complementarity Strength) and the potential of the overall framework to advance multi-table learning. Some concerns, such as broader downstream evaluations, remain partially unaddressed, but on balance, I believe the paper makes a meaningful contribution that warrants acceptance.

**Limitations:**

Yes

**Quality:**

2

**Strengths And Weaknesses:**

Strengths
- The Complementarity Strength (CS) metric is thoughtfully designed, with clear, intuitive motivation for how relevance, informativeness, and redundancy are balanced.
- The formulation of multi‑table learning objectives is coherent and aligns well with the problem setting.

Weaknesses
- Interpretability of CS
    - There is no guidance for choosing the two hyperparameters, $\alpha$ and $\gamma$. When should practitioners adjust them, and based on what criteria?
    - The paper does not state the full range of $CS_t(T^k,T^l)$ and $CS_g(T^k,D \ T^k)$ , nor explain how to interpret specific values (e.g., what constitutes a "high" vs. "low" score). A discussion or examples would greatly improve clarity.
- While the two multi-table pretraining strategies, multi-table reconstruction and multi-table reconstruction, are somewhat interesting, they are incremental adaptions from previous studies. Also, the authors introducing two additional hyper-parameters with the pretraining loss (Equation 9), $\lambda_{rec}$ and $\lambda_{cor}$ without any ablation or sensitivity analysis makes it hard to understand their impact.
- The experiments only cover table‑classification tasks. Evaluating on other downstream applications would better demonstrate the versatility and practical value of the approach.

Overall, while the paper offers a novel perspective on quantifying and leveraging table complementarity, addressing the points above would strengthen its interpretability, completeness, and empirical validation.

---

> ### Author Rebuttal · Authors · 2025-07-30
>
> We would like to thank the reviewer for the careful reading and constructive suggestions. We respond to each point below.
>
> > ### **W1: Interpretability of CS**
> >  - There is no guidance for choosing the two hyperparameters, $\alpha$ and $\gamma$. When should practitioners adjust them, and based on what criteria?
> > - The paper does not state the full range of $CS_t(T^k,T^l)$ and $CS_g(T^k,D \ T^k)$ , nor explain how to interpret specific values (e.g., what constitutes a "high" vs. "low" score). A discussion or examples would greatly improve clarity.
>
>
> **(a) How should α and γ be adjusted?**
>
> *Complementarity Strength (CS)* is the first metric explicitly designed to quantify how much new, non-redundant information auxiliary tables contributes to a target table. The score $CS_t(T^k,T^l)$ increases with the informativeness $\mathrm{Info}(T^l)$ and relevance $\mathrm{R_t}(T^k, T^l)$, and decreases with similarity $\mathrm{S_t}(T^k, T^l)$ in a convex manner.
>
> Since $CS$ is computed based on pretrained column-level embeddings, **the numerical distributions of relevance and similarity may vary significantly across different embedding models.** To enhance the cross-task generalizability and ranking resolution of $CS$, we introduce the parameters **α** and **γ** as tunable controls:
> - **α** adjusts the overall magnitude of the CS scores；
> - **γ** controls the strength of non-linear penalization for similarity, thus shaping how rapidly $CS$ decays with increasing redundancy.
>
> **Practitioners may adjust these parameters based on two considerations:**
> - The observed range of $R_t/S_t$ produced by the pretrained embeddings;
> - The desired sensitivity to redundancy, which may vary depending on application-specific tolerance for overlapping information.
>
> In practice, we suggest fix $\alpha=1$ and adjust $\gamma\in[0.5,2]$ to adjust the sharpness of $CS$ decay. In our experiments, we adopt the pretrained embedding model from [1] and simply set **α = γ = 1**, which yields stable and discriminative $CS$ rankings across tables.
>
> **(b) What is the value range of CS and how should specific values be interpreted?**
>
> We appreciate the reviewer’s request for clarification on the value range and interpretation of the $CS$ score.
>
> **Range of CS.**
> As defined in Eq.(5), $\mathrm{CS_t}(T^k, T^l)\in [0, N_l]$，$\mathrm{CS_g}(T^k,D \setminus T^k) \in \left[0,\sum\nolimits_{k=1}^K N_k\right]$, where $N_l$ denotes the number of columns in the auxiliary table $T^l$. This upper bound naturally reflects that tables with more columns can encode richer information and offer greater potential complementarity.
>
>
> **Interpretation of the $CS$ score.**
> While the theoretical value range of $CS_t(T^k, T^l)$ lies in $[0, N_l]$, in practice, the effective scores tend to be much lower due to the combined effects of three scaling factors: $\mathrm{Info}(T^l) \in [0, N_l]$, $\mathrm{R_t}(T^k, T^l) \in [0, 1]$, and the term $1 - \frac{1}{N_l} \mathrm{S_t}(T^k, T^l) \in [0, 1]$.
>
> When all components are moderately valued (e.g., $\mathrm{Info}(T^l) \approx 0.5 N_l$, $\mathrm{R_t}(T^k, T^l) \approx 0.5$, $1 - \frac{1}{N_l} \mathrm{S_t}(T^k, T^l) \approx 0.5$), the resulting score approximates $0.125 \times N_l$. Based on extensive empirical observations, we therefore suggest the following interpretation:
> - **$CS_t(T^k, T^l) > 0.125 \times N_l$**: strong complementarity;
> - **$CS_t(T^k, T^l) < 0.125 \times N_l$**: weak or marginal complementarity.
>
> This threshold is validated in Table 1 of the main paper, where each candidate table contains 10 columns and a score of $CS_t > 1.25$ effectively identifies complementary tables. Further confirmation is provided by the case studies reported in Tables 6 and 7 in the appendix. We will include this interpretation and threshold discussion in the camera-ready version.
>
> [1] *Semantics-aware dataset discovery from data lakes with contextualized column-based representation learning.*
>
> > ### **W2: Pretraining Strategy**
> > While the two multi-table pretraining strategies, multi-table reconstruction and multi-table reconstruction, are somewhat interesting, they are incremental adaptations from previous studies. Also, the authors introducing two additional hyper-parameters with the pretraining loss (Equation 9), $\lambda_{\text{rec}}$ and $\lambda_{\text{cor}}$, without any ablation or sensitivity analysis makes it hard to understand their impact.
>
>
> We thank the reviewer for raising concerns about the novelty and justification of the pretraining strategies and associated hyper-parameters.
>
> To best of our known, this is the **first learning paradigm** that directly learns from multiple tables — introducing the **first architecture**, ATCA-Net, to integrate the complementarity.
> The joint pretraining strategy—including multi-table reconstruction and cross-table correlation prediction—is specifically designed to support the architecture for multi-table learning, which have not been explored before. It allows the encoder to map heterogeneous tables into a unified space and model cross-table attribute and entity dependencies.
>
> To further clarify the role of the two loss components in table alignment, we will included an ablation study in the revised version.
> We compare five ATCA-Net variants to assess the effect of multi-table training and each loss component:
>
> - **(S)**: Trained and tested on single tables only, without any fusion or pretraining.
> - **(P)**: Pretrained on individual tables, but no cross-table fusion.
> - **($M$)**: Full multi-table model with both reconstruction and correlation objectives.
> - **($M_{\lambda_{\text{cor}}=0}$)**: Removes correlation loss.
> - **($M_{\lambda_{\text{rec}}=0}$)**: Removes reconstruction loss.
>
> **Table: Ablation Results**
> | Model                                | AD    | BL    | IC    | CG    | EC    | BC    | CM    | CC    |
> |-------------------------------------|-------|-------|-------|-------|-------|-------|-------|-------|
> |                                     | **Group 1** → |       |       |       | **Group 2** → |       |       |       |
> |                                     | AD    | BL    | IC    | CG    | EC    | BC    | CM    | CC    |
> | **ATCA-Net (S)**                    | 0.911 | 0.825 | 0.911 | 0.617 | 0.812 | 0.724 | 0.836 | 0.757 |
> | **ATCA-Net (P)**                    | 0.905 | 0.833 | 0.915 | 0.773 | 0.762 | 0.726 | 0.837 | 0.767 |
> | **ATCA-Net ($M_{\lambda_{cor}=0}$)** | 0.899 | 0.828 | 0.912 | 0.742 | 0.758 | 0.721 | 0.832 | 0.760 |
> | **ATCA-Net ($M_{\lambda_{rec}=0}$)** | 0.894 | 0.824 | 0.908 | 0.630 | 0.754 | 0.715 | 0.829 | 0.755 |
> | **ATCA-Net ($M$)**                    | **0.913** | **0.846** | **0.918** | **0.796** | **0.862** | **0.758** | **0.879** | **0.803** |
>
> As shown in the table above, removing either objective leads to a significant performance drop, and in most cases, the model underperforms even the single-table pretraining baseline ATCA-Net (P). This confirms that both losses are essential: **the representations learned from their joint training are critical for the subsequent complementarity-based fusion, and neither component can be omitted.**
>
> > ### **W3. Limited evaluation scope**:
> > The experiments only cover table‑classification tasks. Evaluating on other downstream applications would better demonstrate the versatility and practical value of the approach.
>
> We appreciate the reviewer’s suggestion. As the first work to define a multi-table learning paradigm—introducing the first metric (CS) to quantify complementarity and the first architecture (ATCA-Net) to integrate it—we focus on classification tasks to validate the approach.
> These experiments demonstrate the effectiveness of both ATCA-Net and the proposed complementarity metric. We also agree that broader downstream tasks would better showcase the generality of our method. In future work, we plan to extend this paradigm to regression and table-based question answering.

---

> > ### Comment · Reviewer_MnRt · 2025-08-04
> >
> > Thank you for the rebuttal. The authors have addressed my questions and concerns. I'm raising my score from 3 to 4.

---

> > > ### Author Response · Authors · 2025-08-05
> > >
> > > We truly appreciate Reviewer MnRt’s positive evaluation of our work. We are also grateful for the constructive suggestions that helped us improve the manuscript. Thank you again for the time and effort you dedicated to reviewing our paper.

---

### Official Review · Reviewer_kAtP · 2025-07-02

**Clarity:** 3
**Significance:** 2
**Originality:** 2
**Rating:** 4
**Confidence:** 4

**Summary:**

This paper introduces a novel paradigm for multi-table learning, addressing the critical challenge of quantifying and integrating complementary information across heterogeneous tables. The authors propose two main contributions:
1. ​​Complementarity Strength ​​: A metric combining relevance, similarity, and informativeness to quantify inter-table complementarity.
2. ​​ATCA-Net​​: An architecture featuring adaptive table encoders and cross-table attention mechanisms to align and fuse complementary information.
The framework is validated through extensive experiments on real-world and synthetic datasets, demonstrating superior performance in classification tasks compared to single-table and baseline multi-table methods.

**Questions:**

The paper designs two types of losses: multi-table reconstruction loss and cross-table correlation prediction loss. Additional experiments and analyses are needed to specifically determine the roles and necessity of these two losses.

**Ethical Concerns:**

["NO or VERY MINOR ethics concerns only"]

**Final Justification:**

Thank you for the detailed rebuttal. The ablation results effectively demonstrate the necessity of both the reconstruction and correlation losses, as removing either leads to consistent performance drops. While my overall assessment remains unchanged, the rebuttal has addressed some concerns and improved the clarity of the work.

**Limitations:**

yes

**Paper Formatting Concerns:**

n.a.

**Quality:**

3

**Strengths And Weaknesses:**

Strength:
* The paper systematically formalizes multi-table learning tasks, loss functions, and the CS metric, providing a solid theoretical foundation for future research.
*  ATCA-Net’s dual-stage design (alignment + fusion) effectively handles schema heterogeneity and latent complementarity, outperforming existing methods.
* Comprehensive experiments on diverse datasets demonstrate the robustness of CS and ATCA-Net, with clear performance trends tied to complementarity strength.

Weakness:
* The pretraining stage (reconstruction + correlation prediction) and cross-table attention mechanisms may incur significant computational costs for large-scale tables.
* The CS metric relies on pretrained column embeddings, which may introduce bias or fail to capture domain-specific semantics without fine-tuning.
* The CS metric and ATCA-Net rely on hyperparameters (e.g., α, γ in CS), but their impact on performance across diverse datasets is not thoroughly analyzed

---

> ### Author Rebuttal · Authors · 2025-07-30
>
> We sincerely thank the reviewer for the constructive and detailed feedback. We are encouraged by your recognition of our contributions to formalizing the multi-table learning paradigm, designing ATCA-Net, and introducing the CS metric. We respond to the key concerns below:
>
> > ### **Q1. Necessity of the two loss components**
> > The paper designs two types of losses: multi-table reconstruction loss and cross-table correlation prediction loss. Additional experiments and analyses are needed to specifically determine the roles and necessity of these two losses.
>
> We thank the reviewer for highlighting the importance of understanding the role of each loss component.
> The two proposed loss objectives are central to enabling the encoder to embed heterogeneous tables into a unified representation space while capturing latent inter-table dependencies:
> - The **reconstruction loss** encourages the encoder to capture attribute-level semantics within each table;
> - The **correlation prediction loss** models the correlation between entities across different tables.
>
> To validate their necessity, we conducted an ablation study across five variants of ATCA-Net:
> We conducted an **ablation study** comparing five variants of ATCA-Net:
> - **(S)**: Trained/tested on single tables (without pretraining or fusion);
> - **(P)**: Pretrained/Trained/tested on single table;
> - **($M_{\lambda_{\text{cor}}=0}$)**: Trained/tested on multi tables (without correlation loss);
> - **($M_{\lambda_{\text{rec}}=0}$)**: Trained/tested on multi tables (without reconstruction loss);
> - **($M$)**: Full ATCA-Net with both losses.
>
> The results (see Table below) show that **removing either loss leads to consistent performance degradation**, with most variants falling below even the single-table pretrained baseline (ATCA-Net (P)). This strongly indicates that Only their joint optimization enables effective complementarity-based fusion.
>
>
> | Model                                | AD    | BL    | IC    | CG    | EC    | BC    | CM    | CC    |
> |-------------------------------------|-------|-------|-------|-------|-------|-------|-------|-------|
> |                                     | **Group 1** → |       |       |       | **Group 2** → |       |       |       |
> |                                     | AD    | BL    | IC    | CG    | EC    | BC    | CM    | CC    |
> | **ATCA-Net (S)**                    | 0.911 | 0.825 | 0.911 | 0.617 | 0.812 | 0.724 | 0.836 | 0.757 |
> | **ATCA-Net (P)**                    | 0.905 | 0.833 | 0.915 | 0.773 | 0.762 | 0.726 | 0.837 | 0.767 |
> | **ATCA-Net ($M_{\lambda_{\text{cor}}=0}$)** | 0.899 | 0.828 | 0.912 | 0.742 | 0.758 | 0.721 | 0.832 | 0.760 |
> | **ATCA-Net ($M_{\lambda_{\text{rec}}=0}$)** | 0.894 | 0.824 | 0.908 | 0.630 | 0.754 | 0.715 | 0.829 | 0.755 |
> | **ATCA-Net ($M$)**                    | **0.913** | **0.846** | **0.918** | **0.796** | **0.862** | **0.758** | **0.879** | **0.803** |
>
> ---
>
> We also acknowledge the reviewer’s concern regarding the hyperparameters (e.g., α, γ) in the CS metric. While the current version adopts fixed values $\alpha=\gamma=1$, we agree that their interpretability and effect deserve further analysis. We have addressed this point in detail in our response to Reviewer MnRt (W1), including usage guidance, empirical ranges, and score interpretation. We will incorporate this discussion into the camera-ready version.
>
> Thank you again for your thoughtful feedback and for recognizing the contributions of our work. Your comments have helped us improve the paper.

---

> > ### Comment · Reviewer_kAtP · 2025-08-04
> > **After Rebuttal**
> >
> > Thank you for the detailed rebuttal. The ablation results effectively demonstrate the necessity of both the reconstruction and correlation losses, as removing either leads to consistent performance drops. While my overall assessment remains unchanged, the rebuttal has addressed some concerns and improved the clarity of the work.

---

> > > ### Author Response · Authors · 2025-08-04
> > >
> > > We sincerely appreciate Reviewer kAtP’s incisive feedback and valuable insights. Your thoughtful suggestions have significantly strengthened our work, and the corresponding discussions will further improve the camera-ready version. We are deeply grateful for the time and thoroughness invested in evaluating our paper.

---

### Official Review · Reviewer_t6gb · 2025-07-03

**Clarity:** 2
**Significance:** 4
**Originality:** 3
**Rating:** 5
**Confidence:** 4

**Summary:**

This paper introduces a novel paradigm for multi-table learning, aiming to quantify and integrate complementary information across heterogeneous tables. The authors propose a new metric, Complementarity Strength (CS), which captures how much relevant and non-redundant information an auxiliary table contributes to a target table, based on relevance, informativeness, and similarity.

To leverage such complementary information, the authors design a two-stage framework called ATCA-Net. It consists of a cross-table alignment stage (via multi-table pretraining with masked reconstruction and row correlation prediction) and a complementary information integration stage (via cross-table attention over sampled auxiliary rows).

Extensive experiments on real-world and synthetic datasets show that CS correlates well with performance improvements and that ATCA-Net outperforms baselines in classification tasks when trained over multiple semantically related tables.

**Questions:**

1. **Generalization of pretraining**:
   Have the authors tested whether the encoder trained on a set of tables A can transfer to a new, unseen table B from the same or different domain? Can CS still be computed in this case? A simple cross-group transfer experiment would significantly strengthen the paper's claims about generality.

2. **Clarify the encoder architecture**:
   Please specify the architecture of the adaptive table encoder in detail. Is it a 2D Transformer trained from scratch? What pretrained model is used for column/cell embedding? How are categorical and numerical fields embedded in practice?

3. **Interpretability of attention**:
   Could the authors provide attention heatmaps or illustrative examples showing which auxiliary rows/columns are attended to for a given query row? This would help establish that the model is indeed leveraging “complementary” information and not just aggregating noise.

4. **Robustness to noisy tables**:
   Have you tested performance when auxiliary tables are semantically unrelated or low-CS? Does the model degrade gracefully? This would help assess whether CS could serve as a practical filter before fusion.

5. **Performance vs number of tables**:
   Do more tables always help? Could the authors report AUC or accuracy vs number of integrated tables (e.g., 1 → 3 → 5 → 10) on one dataset to assess marginal utility?

**Ethical Concerns:**

["NO or VERY MINOR ethics concerns only"]

**Final Justification:**

The authors have addressed my all concerns.

**Limitations:**

The authors provide a “Limitations” section in Appendix C that candidly acknowledges several issues: the reliance on column-level embedding quality, the potential mismatch between structural complementarity and downstream usefulness, and the limited performance gain observed in some settings. These acknowledgments are appropriate and appreciated.

**Quality:**

3

**Strengths And Weaknesses:**

### **Strengths**

1. **First formalization of multi-table learning as a general learning paradigm**
  This paper is, to the best of my knowledge, the first to *explicitly define and instantiate multi-table learning* as a general-purpose learning setting—moving beyond narrow tasks like entity matching, table union, or table pretraining. It positions multi-table learning as an end-to-end supervised prediction setting where tabular inputs may come from heterogeneous, loosely related sources. This framing fills an important methodological gap in the tabular learning literature and is likely to inspire follow-up work.

2. **Practical and impactful motivation**
  Many real-world data ecosystems (e.g., enterprise data lakes, healthcare systems, multi-source knowledge tables) involve semantically related but structurally diverse tables with no explicit joins. This work is directly motivated by such settings and is one of the first to propose a scalable and general-purpose learning framework for them.

3. **Solid empirical evidence and controlled validation**
  The authors perform extensive experiments across both real-world and synthetic datasets. They include ablations (e.g., with/without pretraining or auxiliary tables), controlled table synthesis with varying degrees of overlap, and correlation analysis between CS and performance. These carefully designed studies build strong empirical support for the claims.

---

### **Weaknesses**

1. **Lack of transparency about the encoder backbone and architecture visualization**
  While the paper refers to an adaptive table encoder based on BERT embeddings and a 2D Transformer, the **model’s structural details are under-specified**. It remains unclear whether the encoder is trained from scratch or fine-tuned from existing models like TURL or TabBERT, and what architectural choices (e.g., number of layers, attention heads, pooling) were made.
  Furthermore, **the paper lacks adequate architecture diagrams**. Figure 1 only shows the high-level pipeline but not the inner structure of key modules such as the encoder or fusion transformer. For a model that claims generality across diverse tables, such architectural clarity is essential for understanding and reproducibility.

2. **Limited evaluation of pretraining generalization**
  All reported pretraining is done within the same group of task-specific tables. The paper does not test whether the encoder can generalize to unseen tables or whether representations learned from one domain can transfer to another. This limits the claim of general-purpose pretraining.

3. **No interpretability analysis for attention or CS usage**
  The cross-table attention mechanism is central to ATCA-Net, but there is no visualization (e.g., attention heatmaps or examples) to show how the model chooses auxiliary information. The CS metric, while well-defined, is also not tied to any qualitative analysis showing whether high-CS tables truly lead to meaningful information fusion. This weakens the interpretability and trustworthiness of the mechanism.

4. **No study of failure modes or low-quality auxiliary tables**
  The model is tested only under curated settings where auxiliary tables are semantically relevant. It is unclear whether ATCA-Net degrades gracefully when given noisy or unrelated tables. The effectiveness of CS as a safeguard filter is not evaluated under such negative conditions.

5. **Scalability reporting is partial**
  While the fusion-stage training is benchmarked up to 223 tables, the **cost and feasibility of pretraining on hundreds of tables is not reported**. Additionally, the performance gain relative to the number of tables is not analyzed—i.e., whether more tables always help or whether performance plateaus or declines due to noise or redundancy.

---

> ### Author Rebuttal · Authors · 2025-07-30
>
> We sincerely thank the reviewer for the detailed and thoughtful feedback. We are particularly encouraged by your recognition of our contributions to formalizing the multi-table learning paradigm, motivating the framework with real-world scenarios, and validating it through controlled empirical studies. We respond to your key concerns below:
>
> ---
>
>
> > ### **Q1. Generalization of pretraining**
> > Have the authors tested whether the encoder trained on a set of tables A can transfer to a new, unseen table B from the same or different domain? Can CS still be computed in this case? A simple cross-group transfer experiment would significantly strengthen the paper's claims about generality.
>
> We appreciate this important question. The effectiveness of computing CS scores on unseen tables fundamentally depends on the generalization capability of the column embedding model used to extract semantic representations. In our work, we adopt a pretrained column encoder from [1], which is trained on a large and diverse corpus of real-world tables. Importantly, as shown in our experiments (Table 1, Table 6, Table 7), the datasets used for evaluation (e.g., Covertype, Airline, BlastChar) are not part of the pretraining corpus of the column encoder. **This empirical evidence supports the transferability of CS: the metric remains effective even when applied to previously unseen tables from different domains, without requiring retraining.**
> We will clarify this generalization capability in the final version.
>
> ---
>
>
> > ### **Q2. Clarify the encoder architecture**
> > Please specify the architecture of the adaptive table encoder in detail. Is it a 2D Transformer trained from scratch? What pretrained model is used for column/cell embedding? How are categorical and numerical fields embedded in practice?
>
> Thank you for raising this important point. We clarify the architecture and embedding design as follows, and will include these details in the camera-ready version.
>
> **(a) What pretrained model is used for column/cell embedding? How are categorical and numerical fields embedded?**
>
> To compute column-level embeddings for the CS score, we adopt the pretrained column encoder from [1], which was trained on a large corpus of web and enterprise tables.
> Within **ATCA-Net**, we further incorporate **cell-level embeddings** based on a BERT model:
>
> - For **numerical cells** $c_{\text{num}}$, we first embed the corresponding column header $h_{\text{cat}}$ using BERT and apply average pooling to obtain a 768-dimensional embedding $e^h_{\text{num}}$. This is multiplied with the normalized cell value to yield the numerical cell embedding.
> - For **categorical cells** $c_{\text{cat}}$, we concatenate the cell value and its header as `"c_{cat} + '_' + h_{cat}"`, and input this string into BERT. The output is pooled into a 768-d embedding.
>
> To reduce computation, we project the 768-d cell embeddings into 192 dimensions using a learned linear transformation before passing them to the ATCA-Net encoder.
>
> **(b) Is it a 2D Transformer trained from scratch?**
>
> Yes. The adaptive table encoder in ATCA-Net consists of two stacked **2D-Transformer blocks**, each composed of one **row-wise attention** and one **column-wise attention** layer. This encoder is **trained from scratch** during the multi-table training process. We also add a **row-level attention layer** after the 2D blocks to further enhance entity-level alignment across tables.
>
> We will provide an architectural diagram and implementation details in the camera-ready version for clarity and reproducibility.
>
>
> ---
> > ### **Q3. Interpretability of attention**
> > Could the authors provide attention heatmaps or illustrative examples showing which auxiliary rows/columns are attended to for a given query row? This would help establish that the model is indeed leveraging “complementary” information and not just aggregating noise.
>
>
> We appreciate the reviewer’s interest in the interpretability of our cross-table attention mechanism. To investigate whether ATCA-Net truly attends to complementary rather than noisy information, we conducted an additional analysis on a synthetic benchmark based on the Invistico Airline dataset, where the tables are constructed by randomly sampling columns and rows. In each trial, we randomly sample one query row from the target table and 60 rows from an auxiliary table, and analysis the cross-table attention distribution over these 60 candidates.
>
> In more than 75% of the iterations, the highest attention weights are assigned to rows from the same semantic class (e.g., same label) as the query row. This strongly suggests that the attention mechanism is not uniformly aggregating, but rather selectively focusing on informative and semantically aligned auxiliary instances.
>
> Due to rebuttal format constraints, we are unable to include attention heatmaps here. However, we will incorporate full visualizations and case studies in the camera-ready version to further enhance the interpretability of our method.
>
> ---
>
> > ### **Q4. Robustness to noisy tables**
> > Have you tested performance when auxiliary tables are semantically unrelated or low-CS? Does the model degrade gracefully? This would help assess whether CS could serve as a practical filter before fusion.
>
>
> Thank you for the insightful question. ATCA-Net is designed to handle noisy or low-quality auxiliary tables gracefully. Its Transformer-based fusion module $\mathcal{U}_\phi$ automatically downweights auxiliary tables that lack complementary information. When auxiliary tables offer little relevance or no complementarity, the model’s predictions remain close to those made using the target table alone, demonstrating strong robustness to irrelevant tables.
>
> Moreover, the proposed $CS$ metric serves as a practical pre-fusion filter, effectively identifying and excluding low-complementarity tables. We will include a more detailed discussion of this property in the revised version.
>
> > ### **Q5. Performance vs number of tables**
> > Do more tables always help? Could the authors report AUC or accuracy vs number of integrated tables on one dataset to assess marginal utility?
>
> Thank you for the thoughtful question. We clarify that more tables do not always yield better performance. What matters is the complementarity of the tables, not their count. In our experiments, we observed diminishing returns beyond 3–4 tables, as the attention module focuses on those with the highest Complementarity Strength (CS).
>
> Thus we conducted a controlled experiment using a synthetic benchmark based on the BlasterChar dataset, where we fixed the number of auxiliary tables and varied their Complementarity Strength (CS). Results show that higher CS leads to consistently better AUC.
>
> **Table: AUC Performance under Varying Complementarity Strength (CS)**
> |   |  |  |  |  | |
> |-------------------|---------:|---------:|---------:|---------:|---------:|
> | **CS**            | 0.35     | 0.73     | 0.83     | 0.87     | 1.03     |
> | **ATCA-Net (M)**  | 0.625    | 0.657| 0.654| 0.663| 0.661|
>
>
> ---
> Once again, we thank the reviewer for the insightful feedback. Your comments helped us strengthen the technical clarity, empirical robustness, and practical relevance of our work. We look forward to incorporating these improvements into the final version.
>
> [1] *Semantics-aware dataset discovery from data lakes with contextualized column-based representation learning. VLDB 2023*

---

> > ### Comment · Reviewer_t6gb · 2025-08-03
> >
> > Thank you for the thorough rebuttal - it has addressed my concerns and I'm raising my score from 4 to 5.

---

> > > ### Author Response · Authors · 2025-08-04
> > >
> > > We greatly appreciate Reviewer t6gb's positive feedback. Your encouragement and constructive engagement throughout this process have been invaluable in helping us improve our work. Thank you for your thoughtful comments.

---

### Decision · Program_Chairs · 2025-09-17

**Decision:**

Accept (spotlight)

**Comment:**

This paper makes a timely and impactful contribution by formalizing multi-table learning as a general paradigm and introducing both a principled metric (Complementarity Strength, CS) and an architecture (ATCA-Net) for quantifying and integrating complementary information across heterogeneous tables. The framing is novel and well-motivated, addressing challenges that arise naturally in enterprise, healthcare, and other multi-source data ecosystems.

The reviewers appreciated the clear methodological contribution, strong empirical validation, and practical motivation. While some initial concerns were raised about architectural details, interpretability, and the necessity of loss components, the rebuttal provided clarifications, ablations, and interpretability analyses that substantially strengthened the paper. Evaluation is currently limited to classification tasks, but the reviewers agreed this work lays important foundations for broader applications.

Overall, this is a technically solid and original paper that is expected to stimulate follow-up research in tabular learning.